# Race, Social Context, and Caregiving Intensity: Impact on Depressive Symptoms Among Spousal Caregivers

**DOI:** 10.3390/ijerph22091379

**Published:** 2025-09-03

**Authors:** Florence U. Johnson, Melissa Plegue, Namratha Boddakayala, Sheria G. Robinson-Lane

**Affiliations:** 1School of Nursing, University of Michigan, Ann Arbor, MI 48109, USA; grices@umich.edu; 2School of Medicine, University of Michigan, Ann Arbor, MI 48109, USA; petrelim@umich.edu; 3School of Information, University of Michigan, Ann Arbor, MI 48109, USA; namrthdb@umich.edu

**Keywords:** depression, caregivers, Black or African American, mental health, resilience, psychological, race factors, protective factors

## Abstract

Understanding the factors that influence the mental health of family caregivers is essential. This study examines the impact of caregiving intensity, operationalized as caregiving hours, on depressive symptoms, with a focus on racial differences and gender effects. We analyzed data from n = 2112 unique spousal caregivers across 6622 person-years of observations from the Health and Retirement Study (HRS) longitudinal data (2008–2014). We estimated the impact of caregiving hours on depressive symptoms, controlling for race, gender, and education. Random intercepts accounted for household-level variance. We assessed interaction terms to evaluate differential effects across racial groups. Depressive symptoms were positively associated with caregiving hours β = 1.74, SE = 0.24, suggesting that increasing caregiving hours is likely to lead to emotional distress. We observed a significant negative interaction effect among African American caregivers (β = −1.25, *p* = 0.013), indicating that increased caregiving hours led to a reduction in depressive symptoms. Gender was not significantly associated with caregiving hours (β = −0.36, *p* = 0.715). The random effects model demonstrated substantial household-level variation (var(_cons) = 266.07, *p* < 0.0001). Findings highlight racial differences in the effects of caregiving hours on depressive symptoms and point to the importance of culturally responsive interventions designed to mitigate depressive symptoms among caregivers. Future research should explore protective factors that mitigate psychological distress and promote resilience.

## 1. Introduction

More than 11 million unpaid family caregivers in the United States provided nearly 18.4 billion hours of care to those living with dementia in their homes [1]. Research shows that around 10% of the caregivers provide care to their spouses with Alzheimer’s Disease and Related Dementias (ADRD) [2], and 30% of the ADRD caregivers are 65 or older [1]. As Baby Boomers continue to age, we expect the number of people living with dementia to increase, which will intensify the dependence on spousal caregivers [3]. This shifting of the demographic highlights the importance of grasping and confronting the mental health issues associated with spousal caregiving, especially for Black spousal caregivers [3]. These caregivers tend to provide care for more extended periods compared to those caring for individuals with other chronic conditions [1]. Spousal caregivers of individuals with dementia frequently face significant challenges that negatively affect their mental well-being and the quality of care they provide [3,4]. These caregivers often lack training in essential disease management skills, such as assisting with activities of daily living or managing behavior issues, which contributes to the challenges of managing the complex care tasks associated with dementia [5,6]. The emotional demands of caregiving, which include tasks such as assisting with mobility and personal hygiene, can be overwhelming, particularly for older caregivers who may have health challenges of their own [5,7,8,9]. 

Although caregiving can be meaningful, research has found it to be associated with significant psychosocial and emotional strain, placing the caregiver at risk for experiencing depressive symptoms, anxiety, and a decline in overall well-being [10,11,12]. Mental health challenges are especially prevalent in the Black and Hispanic communities; however, the associated mental health-related issues in these groups remain understudied [13]. Building on previous research, this study focuses on the contextual factors that are available in the HRS dataset and can impact caregiving, such as gender, education level, and ethno-racial identity [14,15]. These factors are associated with health inequities—i.e., avoidable and unfair differences in health status, access to healthcare, and health outcomes [16]. For Black and Hispanic caregivers, these inequities add stress to their already heavy caregiving loads [17]. Systemic factors, including racism, socioeconomic inequality, and limited access to supportive resources, may exacerbate the stress of caregiving [18,19].

Medical expenses and the need for supplementary paid support services contribute to financial strain among caregivers with limited resources [10,11,12,13,14,15,16,17,18,19,20,21,22]. Gender also plays a crucial role, with women, who tend to be primary caregivers, typically experiencing higher caregiving burden and associated mental health challenges due to societal norms and expectations for prioritizing familial needs over their own and being the backbone of the family [23,24]. Education levels can impact caregivers’ ability to navigate healthcare systems and access resources, with higher education often correlating with better coping strategies and advocacy skills [25,26]. Access-related challenges exacerbate depressive symptoms and stress [17,27].

Despite the growing body of research on caregiver burden and mental health outcomes [20,23], the unique experiences of Black and Hispanic spousal caregivers remain understudied [17,28,29], leaving critical gaps in our understanding of how to support their mental health with responsive cultural interventions that address the stressors commonly faced by Black and Hispanic spousal caregivers [17,18]. One final gap in the literature, given the variation in caregiving intensity across groups, is the relationship between caregiving hours and depressive symptoms [30,31]. Using the Health and Retirement Study (HRS) longitudinal data, we examined the impact of caregiving intensity and depressive symptoms on spousal caregivers. To gain a deeper understanding of caregiver mental health disparities, we stratified the analysis by gender and race [17,21,23].

We hypothesize that as caregiving hours increase, self-reported depressive symptoms will increase, and there will be differences based on race and ethnicity. We expect that Black spousal caregivers will verbalize higher levels of depressive symptoms as their caregiving hours increase. Our overarching goal is to leverage a nationally representative dataset to improve our understanding of racial differences in spousal caregiving and their relationship to the mental health of caregivers.

## 2. Materials and Methods

### 2.1. Data

The data used are from the Health and Retirement Study (HRS). HRS is a nationally representative longitudinal survey of noninstitutionalized individuals aged 50 and older, surveying more than 22,000 people in the United States every two years. If the participant (hereafter referred to as the partner) is married or living with a spouse/partner, the spouse/partner (hereafter referred to as the spouse) is recruited into the study and surveyed, regardless of their age. They used a multistage, area-clustered probability design with an overall response rate of 80% to choose the primary sample [32].

The RAND Health and Retirement Study (RAND HRS) Longitudinal File 2018 is the primary dataset used. It is a streamlined and easy-to-use product from the HRS core and exit interviews [33]. This dataset captures comprehensive sociodemographic, health, and functioning information, including data on households with married or partnered individuals. It encompasses variables such as income, assets, and medical expenditures, facilitating research on retirement, health insurance, savings, and economic well-being [34]. Using Wave 9 (2008–2014) of the RAND HRS for this secondary analysis made it possible to build a substantial longitudinal dataset that tracked each couple member (spouse and partner) over a six-year period [34]. The dataset also included validated measures of cognitive functioning and depressive symptoms [35,36].

### 2.2. HRS Population and Sample Derivation

HRS included a representative sample of older respondents (ages 50 years and older) who met the following inclusion criteria from the original HRS wave 9 study sample: they completed at least two adjacent waves of the HRS from 2000 to 2014 [32]. Wave 9 served as the primary point of reference; the researchers selected participants who consistently participated in the study across all years for the longitudinal data because their participation ensured reliable, longitudinal data [32]. They had a spouse or partner (of any age) who also responded to the two or more adjacent surveys. HRS defined people who assisted their partners with at least one activity of daily living (ADL) or instrumental activity of daily living (iADL) task as spousal caregivers, and their partners also recognized them as caregivers. The collection of demographic data occurred during the first survey [37].

The in-person supplemental study, Aging, Demographics, and Memory Study (ADAMS) gathered demographic information on both study participants’ cognition [38]. The HRS dataset sample includes both members of the spousal dyads; when the study person is diagnosed with ADRD, the person providing care is referred to as the “spousal caregiver”, and the partner with ADRD as the care recipient. Our final analytic sample of 2112 spousal caregivers included participants who contributed to two longitudinal observations during the 2008–2014 data period, self-identified as spouses, and assisted their spouse with ADL or iADL.

### 2.3. Outcome (Depressive Symptoms)

The primary outcome of depression in this study was the number of caregiver depressive symptoms, based on the Center for Epidemiologic Studies Depression (CES-D) scale, which is a valid and reliable way to measure depression [39]. The researchers at HRS used CESD-8, a modified version of the CESD scale that contains eight items; the CESD-8 is typically scored on a four-point Likert scale [40]. This study used a dichotomous scoring system for simplicity and ease of interpretation. HRS researchers asked respondents to answer ‘yes’ or ‘no’ to questions related to depressive symptoms. Each ‘yes’ response was coded as ‘one’ and each ‘no’ response as ‘zero,’ except for two positively worded items: ‘I felt cheerful’ and ‘I loved life.’ For these two items, a ‘yes’ response was coded as ‘zero’ and a ‘no’ response as ‘one’ [41].

The total score was calculated by summing the coded responses, with a maximum possible score of 8. HRS also asked the respondents if they experienced feelings of depression, found that everything took effort, had sleep disturbance, felt alone, felt unhappy, and struggled to get going. A total score of 5 or more was indicative of depressive symptoms.

#### 2.3.1. Main Predictors of Interest (Care Intensity)

Our primary predictor of interest, care intensity, was operationalized in this study as the number of hours a week a spouse spends helping the person living with ADRD. HRS calculated caregiving hours by asking the study subject two specific questions: How many days did your spouse assist you last month, and how many hours per day does your spouse help you during those days? The answer could range from 0 to 8 h. The spousal caregiver was asked the same questions about caregiving hours as the study subject. Our study team categorized spousal caregiving hours into none, 1–13 (low), and greater than 13 (high) to facilitate descriptive comparisons of demographics with care intensity.

#### 2.3.2. Covariates

Age, education, gender, and marital status were included as covariates since these factors are associated with both care intensity and depressive symptoms.

#### 2.3.3. Analysis

We generated descriptive statistics for all study variables. To comprehensively assess the effects of race, age, education, and caregiving hours on self-reported depressive symptoms, we estimated several models. A linear mixed model was used to examine the influence of gender, education, race, and caregiving hours on depressive symptoms, accounting for repeated measures across time. To explore whether the relationship between caregiving hours and depressive symptoms varied by race, we included an interaction term between caregiving hours and caregiver race. We used maximum likelihood estimation to fit the mixed-effects model, which included a random intercept for each caregiver to account for within-subject correlation due to multiple years of data. Predictive margins were calculated and plotted at varying levels of caregiving hours to visualize the interaction effects and to compare depressive symptom trajectories across racial groups. Data management and analysis were performed using STATA 16.

## 3. Results

We identified an eligible cohort of 2112 unique spouses across 6622 person-years in which they cared for partners identified as having ADRD. The majority of the spouses were non-Hispanic White (NH White) (60.9%), followed by African American (20.1%), Hispanic (11.5%), and other (7.4%); Table 1. Most of the caregivers were female (59.6%), with less than a “High School (HS) education” (36.2%), and fewer spouses had “some college” (17%). Spouses in the “high” caregiving level category exhibited the highest average depressive symptoms (mean = 2.06, standard deviation (SD) = 2.18), while those in the “none” category had the lowest (mean = 1.76, SD = 2.05) (Table 2). The linear mixed model analysis (Table 3) revealed key findings regarding the impact of caregiving hours on depressive symptoms, differentiated by race. NH White caregivers showed a significant positive association between caregiving hours and depressive symptoms, with a marginal effect (dy/dx = 0.00347, *p* < 0.0001). Similarly, Hispanic caregivers also showed a positive association between caregiving hours and depressive symptoms (dy/dx = 0.00316, *p* = 0.021). Findings also demonstrated that as caregiving hours increased, depressive symptoms increased for both NH White and Hispanic caregivers. This suggests that caregiver intensity may function inversely across racial groups. African American caregivers had no significant association (dy/dx = 0.00034, *p* = 0.807) with depressive symptoms; these caregivers did not show a significant association, unlike their counterparts (Figure 1). Figure 2 illustrates the relationship between caregiving hours and depressive symptoms stratified by race, education, and gender.

## 4. Discussion

This study underscores the complex link between depressive symptoms and care intensity, demonstrating substantial racial disparities in caregiving reactions to emotional distress. Although depressive symptoms were generally associated with increased caregiving hours, African American caregivers exhibited a contrasting trend, reporting fewer depressive symptoms as caregiving hours increased. These results suggest that cultural, psychological, and structural factors may influence caregiving impact distinctively across ethnic groups. Other studies have noted that Black women commonly use positive reappraisal as an adaptive coping strategy, which may diminish depressive symptoms [42,43,44].

The emotional toll of caregiving may be more pronounced among NH White and Hispanic caregivers, who exhibit stronger associations between caregiving intensity and depressive symptoms (*p* = 0.021) [45]. In contrast, African Americans and individuals from other racial groups demonstrate a weaker or non-significant association [46]. The absence of significant impacts for African Americans and the “other” race groups could be indicative of caregiving dynamics that differ across cultural contexts [47]. Disparities in caregiving practices, coping methods, and social support networks, as well as larger systemic inequities in healthcare and family dynamics, could explain these differences [48]. The lack of gender differences suggests that both men and women experience similar caregiving demands, which highlights the importance of gender-neutral caregiving policies [49].

In their recent qualitative findings, Robinson-Lane et al. [43] reinforce the idea that Black caregivers frequently draw on culturally rooted resilience strategies, such as adaptability, proactive planning, and reliance on supportive environments. However, the cultural frameworks—exemplified by the strong Black woman schema—may also exacerbate the underreporting of mental health issues, hence complicating the recognition and treatment of caregiver mental health. To lessen the psychological impact of caregiving, future research should look into protective characteristics, especially in groups where emotional distress is already high. Caregiver stress can be better understood by examining the role of resilience mechanisms, including social support, coping mechanisms, and access to mental health resources. To create equitable support networks for diverse caregiving populations and address gaps in caregiving experiences, it is essential to understand these dynamics across racial and cultural contexts. Developing culturally responsive interventions is critical.

## 5. Conclusions

Our study results indicate a link between caregiving hours and depressive symptoms in Hispanic and NH White caregivers, but not among African American caregivers. To effectively support the mental health of spousal caregivers, interventions should be accessible and culturally sensitive to honor the values, coping strategies, and lived experiences of those they aim to serve, especially African American caregivers. Regular mental health screenings should include checks for depressive symptoms, utilizing the resilience-building strategies developed by Black caregiving groups. It should be noted that although our study did not find a significant association between caregiving hours and depressive symptoms among African American caregivers, this does not mean that they might not face unique challenges as caregivers and still need tailored support. Addressing structural inequities, such as limited access to resources and training, financial constraints, and examining the interplay of cultural identity, gender norms that typically assign caregiving responsibilities to women, and disengaged social support networks, is essential for future research and for monitoring resilience and distress over time.

## 6. Limitations

One limitation of this study is that we did not consider participants’ mental health before caregiving because we were interested in how caregiving impacted mental health and not overall mental health status. Finally, bias is possible due to the nature of longitudinal studies, as participants may drop out over time, while new participants are added. Additionally, we addressed missing data using a listwise deletion approach. This limitation may have reduced our sample size, which could have decreased our statistical power and thereby hindered our ability to detect significant associations.

## Figures and Tables

**Figure 1 ijerph-22-01379-f001:**
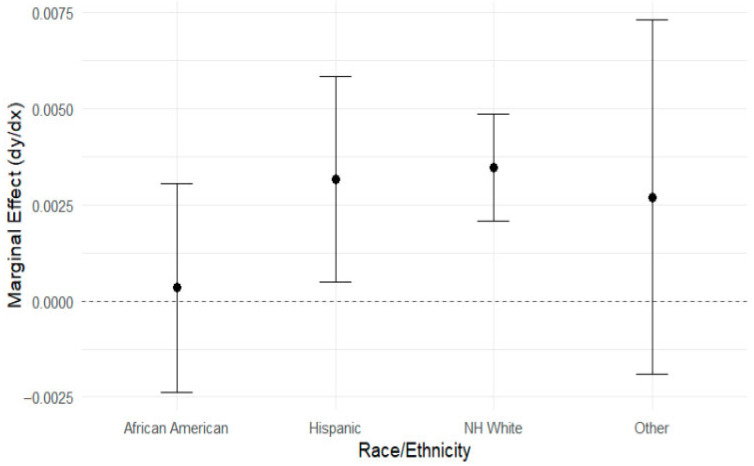
Marginal effects of caregiving hours on depressive symptoms by race. Number of observations: 6622. Expression: linear prediction (fixed portion) with respect to variable cg_sp_wklyhrs (caregiving hours).

**Figure 2 ijerph-22-01379-f002:**
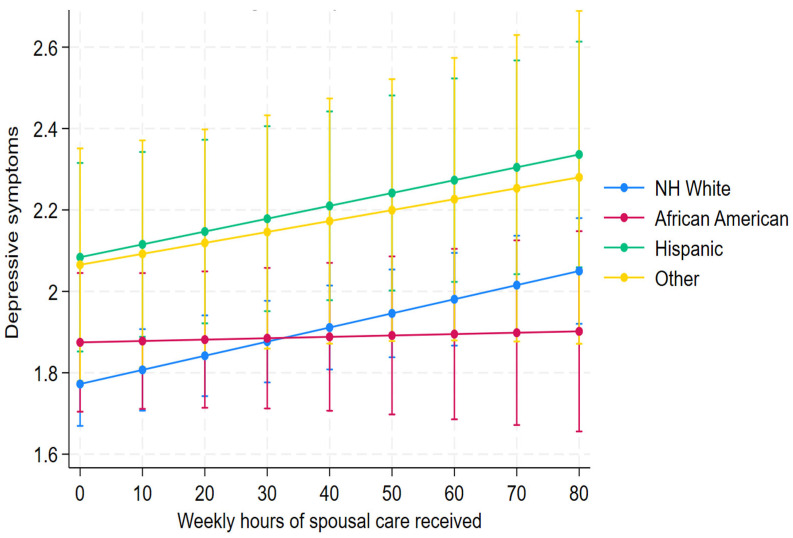
Predictive margins of spousal race with 95% confidence intervals (CIs).

**Table 1 ijerph-22-01379-t001:** Spouse characteristics (n = 2112).

Characteristic	Category	Frequency (n)	Percentage (%)
Race/Ethnicity	NH White	1288	60.98
	African American	425	20.12
	Hispanic	243	11.51
	Other	156	7.39
Sex	Female	1259	59.61
	Male	853	40.39
Education	Less than HS	764	36.17
	GED or HS graduate	757	35.84
	Some college	358	16.95
	College and above	233	11.03
Total		2112	100.00

**Table 2 ijerph-22-01379-t002:** Time-varying characteristics by caregiving level.

Spousal Depression	None (Mean ± SD) n = 4233	Low (Mean ± SD) n = 931	High (Mean ± SD) n = 1458
Depression	1.76 ± 2.05	1.96 ± 2.19	2.06 ± 2.18
ADLs	0.44 ± 1.01	0.43 ± 0.98	0.34 ± 0.83
iADLs	0.36 ± 0.90	0.32 ± 0.83	0.25 ± 0.68
Diagnosis duration (n)	n = 2744	n = 704	n = 1217
Diagnosis duration	2.73 ± 3.78	2.77 ± 3.42	2.97 ± 3.75

**Table 3 ijerph-22-01379-t003:** Linear mixed model results.

Parameter	Estimate	Std. Err.	[95% Conf. Internal]
Caregiving Hours (Race = African American)	−1.2499	0.5053	−2.2403 to −0.2596
Caregiving Hours (Race = NH White)	Ref		
Caregiving Hours (Race = Hispanic)	−0.3004	0.5469	−1.3724 to 0.7716
Caregiving Hours (Race = Other)	−0.3780	0.7127	−1.7749 to 1.0190
Depressive symptoms	2.086863	0.0817672	1.932602 to 2.253438
Gender (Male)	−0.3618	0.9906	−2.3034 to 1.5798
White (Caregiving Hours = 0)	Ref		
Black (Caregiving Hours = 0)	−0.2381	1.5849	−3.3444 to 2.8682
Hispanic (Caregiving Hours = 0)	0.9564	2.0072	−2.9778 to 4.8905
Other (Caregiving Hours = 0)	−1.3794	2.4826	−6.2452 to 3.4863

## Data Availability

The data are publicly available at https://hrsdata.isr.umich.edu/ (accessed on 17 June 2025).

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
