# Peer review of "Race, Social Context, and Caregiving Intensity: Impact on Depressive Symptoms Among Spousal Caregivers"

_ijerph, 2025, doi:10.3390/ijerph22091379_

Round 1
Reviewer 1 Report
Comments and Suggestions for Authors
The approach presented in the article is of interest, as it reflects differences in caregiving with racial distinctions serving as a foundational element. The breadth of the sample and the use of validated, pre-established tools provide a rigorous research framework.
However, there are several issues that would benefit from further clarification:
1. The title refers to Spousal Caregiver Mental Health: Examining the Relationships Between Social Context, Care Intensity, and Depression. However, a key variable throughout the study is race. As the article conducts analyses and draws conclusions with this factor as a central axis, it would be appropriate for the title to reflect this aspect.
2. The inclusion criteria for participants and the longitudinal analyses are described, along with prior considerations. Given that the study analyses correlations between certain variables and depression, the article does not clarify whether participants had, or currently have, any mental health conditions at the time of the study. It is important to consider that a caregiver may already exhibit depressive symptoms due to factors unrelated to caregiving. This point should be clarified.
3. Age is another important factor. Although it is noted that participants are over the age of 50, there are significant differences in capacity and energy for caregiving between individuals in their early 50s and those in their 70s or older. It would be useful to specify the age range covered by the sample.
4. Furthermore, socioeconomic and cultural context may vary significantly. If there are aspects of caregiving that are differentially affected by these parameters, have they not been observed, studied, or taken into account? This point should also be clarified.
5. The conclusions highlight differences and point to the importance of implementing diversified interventions, primarily based on race. It would be advisable to be more explicit about the contributing factors, summarised clearly at the end, to indicate where efforts should be focused.
6. In terms of formatting, the references section should be reviewed. There are some inconsistencies in the use of italics and punctuation.

Author Response
Thank you for your suggestions, and please see the attachment.

Reviewer 2 Report
Comments and Suggestions for Authors
Abstract
The abstract mentions the analysis of "10,120 observations in 2,208 caregiver households between 2008 and 2010", while the results section refers to a longer follow-up period (e.g., "2,221 unique spouses over 6,622 person-years of caregiving"). This inconsistency between the timeframe/sample reported in the abstract and in the rest of the article may cause confusion and should be corrected for consistency. Additionally, the phrasing of some sentences in the abstract may lead to ambiguous interpretations. For instance, it states that "the effects of depressive symptoms on caregiving hours were estimated," followed by "depressive symptoms were positively associated with caregiving hours. This wording inverts the expected direction of the relationship (making it seem as if depression influences caregiving hours). It is advisable to rephrase this to clearly indicate that the primary interest is the impact of caregiving intensity on depressive symptoms (not the reverse). Overall, it is recommended to align the quantitative details in the abstract with those in the methodology and to clarify the direction of the reported association, ensuring that the abstract accurately reflects the article’s content.
Introduction
As a suggestion for improvement, the authors could make the reasoning linking the literature review to the hypothesis more explicit—for example, by emphasizing why, based on the described inequalities, it was expected that Black caregivers would experience a disproportionately greater decline in mental health with higher caregiving loads. Additionally, it is worth noting that the introduction mentions several factors (income, sexual orientation, etc.), but not all were actually included in the analysis. To maintain focus, the authors could limit the initial discussion to the factors later considered in the analysis or briefly justify the decision not to include certain variables (e.g., income is mentioned in the hypothesis but is not listed among the covariates used in the methods).
Methodology
Although the methodological description is detailed, some aspects could be improved in terms of clarity and alignment with other sections of the article. First, it would be helpful to explicitly state the final sample size analyzed (how many caregivers and how many longitudinal observations were included), as this information appears only indirectly in the results. Related to this, the exact period of data used should be clarified: the methodology mentions Wave 9 of the HRS (2008) as the starting point and indicates longitudinal follow-up, but does not make clear up to which year data were analyzed. While the reader may infer that waves up to 2014 were used (as mentioned in the inclusion criteria), the abstract states “between 2008 and 2010”, creating ambiguity.
Second, there is a minor conceptual inconsistency: the hypothesis stated in the introduction suggests controlling for income, but income does not appear in the list of covariates actually used. It is presumed that income was ultimately not included in the model (possibly due to collinearity or missing data), but this was not clarified. It is recommended that the authors justify this decision or revise the formulation of the hypothesis to reflect the covariates that were actually included.
Although the methodology covers variables and analyses well, there is no mention of procedures to control for potential biases (e.g., whether respondents who remained in the study until 2014 differed from those who dropped out, or whether missing data were handled). Given the specialized readership, these details do not necessarily need to be in the methods section but acknowledging methodological limitations later (in the discussion) would be important.
Finally, the HRS is a publicly available and widely used dataset that has received prior ethical approval and includes informed consent from participants. As the authors relied on anonymized secondary data, there is no ethical violation. However, it would be important to provide a clearer justification regarding the ethical considerations, particularly given that the study involves vulnerable populations.
Results
In contrast, “no significant association was observed between caregiving hours and symptoms among Black caregivers (p = 0.807),” suggesting a marked difference in this group. The text also mentions the absence of an effect of caregiver gender ("gender had no significant association") and highlights the presence of unexplained variance between families (significant random intercept variance), although these points are clearer in the abstract than in the main results section. Overall, the writing sticks to reporting what the numbers indicate, without over-interpretation, which is appropriate for the results section.
As an improvement, the authors could consider including brief practical interpretations of the coefficients for readers less familiar with statistical outputs. For example, clarify that a dy/dx of ~0.0035 implies that each additional weekly hour of caregiving is associated with a very small increase (0.0035) in depressive symptoms (on a 0–8 scale), although statistically significant given the large N. This would provide a sense of the effect’s magnitude, complementing its statistical significance.
Another issue to consider is coherence in causal/associative terminology: in the results section, the authors correctly treat the findings as associations (e.g., “caregiving hours were associated with more symptoms”), but as previously mentioned, in the abstract and parts of the discussion, there is some confusion about the direction of the relationship. It would be important to unify the language to make clear that the model examines the effect of caregiving intensity on depressive symptoms (not the other way around). Such unification would strengthen the alignment between the results and the study objective.
Discussion
The absence of a dedicated segment on study limitations was noted, which would be expected in a scientific article to contextualize results cautiously. Including a reflection on limitations would demonstrate rigor and transparency.
There is a small inconsistency that should be corrected: the authors state in the discussion that there was “no significant effect for the Hispanic and ‘Other’ groups”, attributing this to contextual differences. However, as reported in the results, the effect for Hispanic caregivers was significant and positive (p = 0.021). The phrasing here seems to conflict with the actual findings—possibly the authors meant to say no effect for the “Other” group only, or that there was no differential effect for Hispanics. In any case, this sentence should be adjusted to avoid confusion, aligning it with the results (e.g., by acknowledging that a significant effect similar to that of White caregivers was observed among Hispanic caregivers).
Conclusion
Given that the study also found a significant association between caregiving burden and depression among Hispanic and White caregivers, it may be useful to briefly mention broader implications—for example, emphasizing the need for support and interventions for caregivers in general who face high caregiving loads, regardless of group, even if tailored in a culturally appropriate manner. Such an inclusion would ensure that the conclusion encompasses all key findings.
Author Response

(The authors gave the same response as above.)

Round 2
Reviewer 2 Report
Comments and Suggestions for Authors
Overall, the authors have done a good job and the manuscript shows promise. I recommend a thorough language edit for clarity and consistency, a careful review of the reference list to ensure full compliance with the journal’s style (complete metadata, correct DOIs/URLs, consistent citation format), and improved table formatting (uniform titles/captions, clear column labels, units, notes, and alignment).
Author Response
Thank you for your suggestions. We addressed the flow and clarity of the manuscript during this revision.
